# Evaluation of Newcastle Disease antibody titers in backyard poultry in Germany with a vaccination interval of twelve weeks

Björn Oberländer[1]*, Klaus Failing[2], Celina M. Jüngst[1], Nicole Neuhaus[1], Michael Lierz[1], Franca Möller Palau-Ribes[1]

1 Clinic for Birds, Reptiles, Amphibians and Fish, Justus-Liebig-University Gießen, Gießen, Germany, 2 Unit for Biomathematics and Data Processing, Veterinary Faculty, Justus-Liebig-University Gießen, Gießen, Germany

* bjoern.oberlaender@web.de

**Data Availability Statement:** All relevant data are within the manuscript and its Supporting Information files.

## Abstract

Newcastle Disease (ND) is a viral disease spread worldwide with a high impact on economy and animal welfare. Vaccination against Newcastle Disease is one of the main control measures in countries such as Germany with endemic occurrence of Newcastle Disease virus in the free ranging bird population. The German Standing Veterinary Committee on Immunization (StIKo Vet) recommends to revaccinate chickens at intervals of six weeks against Newcastle Disease with attenuated live vaccines via drinking water or spray in line with the SPCs (Summary of Product Characteristics) of current vaccines. However, it is still common practice to revaccinate only every twelve weeks because the SPCs of former vaccines proposed a revaccination after checking the antibody titer which based on practical knowledge was typically sufficient for twelve weeks. The aim of this study was to evaluate if a vaccination interval of twelve weeks against Newcastle Disease under field conditions results in sufficient seroconversion to protect flocks. Antibody titers of 810 blood samples from 27 backyard flocks of chickens were analyzed by ELISA- and HI-tests between 69 and 111 days after vaccination of the flocks with attenuated live vaccines of the ND strain Clone 30. Furthermore, data on the flocks such as breed, sex and age were collected through a questionnaire. In this study a sufficient antibody titer was found in 26 of these flocks. Therefore, a vaccination interval of every twelve weeks with the live vaccines tested is suitable for a vaccination protocol against Newcastle Disease. The lack of seroconversion of one flock also emphasizes the need for regular vaccination monitoring by serological testing and re-evaluation of the vaccination process if needed.

## Introduction

Newcastle Disease (ND) is a viral disease spread worldwide of birds caused by virulent strains of the Newcastle Disease virus (NDV). The virus belongs to the genus *Avulavirus*, family *Paramyxoviridae*, order *Mononegavirales* [1]. It is classified as avian paramyxovirus type 1 (APMV-1). AMPV-1 has a negative-sensed, single stranded, filamentous RNA genome and a

**Funding:** The authors received no specific funding for this work.

**Competing interests:** The authors have declared that no competing interests exist.

glycoprotein and lipid membrane. Different strains of AMPV-1 are classified according to their Intracerebral Pathogenicity Index (ICPI) as apathogenic (ICPI = 0.0), lentogenic (ICPI < 0.7), mesogenic (ICPI 0.7–1.5) or velogenic (ICPI > 1.5). Only mesogenic or velogenic strains induce Newcastle Disease [2]. AMPV-1 is commonly spread by indirect or direct contact with infected birds. Possible sources of Newcastle Disease infections are poultry, pigeons and free-ranging birds [3].

Depending on the Newcastle Disease viral strain and the susceptibility of the host, the virus has a morbidity and mortality of up to 100% [4]. Furthermore, the performance of infected flocks decreases significantly and the eggs of infected animals become thin-shelled. Fowl infected with lentogenic strains of the virus usually show only mild respiratory symptoms, however, infections with velogenic strains lead to a catarrhous inflammation of the mucous membrane and central nervous symptoms such as torticollis and opisthotonos [4–6].

In the last two years there have been multiple Newcastle Disease outbreaks in Belgium, Switzerland, Sweden and Slovakia [7]. The recent outbreaks in Europe usually originated from backyard poultry which then infected commercial poultry flocks [3]. An outbreak in poultry with Newcastle Disease virus affects a country's economy substantially, in particular, since countries with a Newcastle Disease outbreak face restricted trading conditions [8]. In most countries poultry are vaccinated against ND to prevent outbreaks. However, there are also countries such as Switzerland, Ireland, Norway or Sweden in which a vaccination against Newcastle Disease is strictly prohibited [9,10]. Vaccination strategies are usually used in countries in which virulent Newcastle Disease virus strains are endemic or infections with low virulent field strains may have significant economic consequences. The reservoir of mildly virulent field strains are free-ranging birds such as wild pigeons (Columbidae) or water fowl such as ducks (Anatidae) or geese (Anserinae) [11–14].

In Germany, Newcastle Disease is a notifiable disease. Control and prevention are described in the 'Geflügelpest-Verordnung' (2005) which is based on the European directive EWG RL 92/40. If an infection in poultry is detected, every flock within a radius of 1 km will be culled and every flock within a radius of 3 km has to be tested for Newcastle Disease [15]. Furthermore, every single chicken and turkey in Germany has to be vaccinated to build up an immunity against the virus [15]. Commonly vaccines with lentogenic APMV-1 strains, like Hitchner B1 or LaSota, as well as apathogenic strains, such as Ulster or VG/GA, are used to vaccinate poultry [16]. The vaccination can be performed by veterinarians or trained commercial farmers and since the last amendment of the Animal Vaccination Law as of 31 March 2020 also by trained backyard poultry keepers [17]. The vaccination against ND by commercial or non-commercial poultry keepers has to fulfil very strict requirements. The veterinarian supplying the vaccine to the poultry keepers has to request a special permit for the vaccination before the first use and must renew the permit every year. The responsible veterinarian also has to train the breeder with regard to the vaccination technique, verification of the vaccination and side effects of the vaccination. In addition to that, the veterinarian has to check the vaccination success and the flocks have to be supervised regularly, at least every three months [18]. Due to these time-consuming administrative efforts it is hardly worthwhile for the typical small-animal veterinarians to supply backyard poultry keepers with ND-vaccines. Instead, it is more likely that small-animal veterinarians will continue to vaccinate backyard poultry flocks themselves, which is, however, more expensive for the backyard poultry breeder. Due to the high costs, there is the risk that backyard poultry breeders do not have their poultry properly vaccinated. At the same time, there is a rising tendency of keeping small numbers of backyard poultry to produce meat and eggs for self-sufficiency or to "save" layers from slaughter often without knowledge of the applicable laws.

It is common practice in Germany to vaccinate backyard poultry with live attenuated vaccines via drinking water, as it is easier and cheaper than using inactivated vaccines that have to be applied intramuscularly. While there are inactivated vaccines, which maintain immunity for at least eight to twelve months, the manufacturers of all currently available live attenuated Newcastle Disease vaccines in Germany state a duration of immunity of only six weeks. In contrast, in former specifications of product characteristics the revaccination scheme proposed was based on antibody-titer to be controlled by the veterinarian via hemagglutination-inhibition (HI) testing. Based on experiences in the field, it has become common practice to vaccinate poultry only every twelve weeks against Newcastle-Disease [19]. Other studies have shown that revaccinated chickens exhibited protection against ND until 55 weeks of age. These chickens were vaccinated with a live-vaccine through eye-drop once and were revaccinated with a live-vaccine intramuscularly for a second time at the age of 32 or 39 days, respectively [20].

The use of live vaccines induces cell-mediated immunity as well as humoral immunity [21]. Cell-mediated immunity alone is not sufficient by itself to protect chickens against ND, because ND-protection is mainly based on local and systemic antibodies [22]. The number of systemic antibodies depends on the invasiveness of the ND-vaccine strain, i.e. the more invasive the strain, the higher the antibody response. Systemic antibodies can be measured easily and cost efficiently via serology [21]. Thus, it is the most common way to analyze immunity against a virus in a field study. The predominant systemic antibody in chicken blood is IgY, with a serum concentration of 5–10 mg/ml. In addition to IgY, there are also IgM-antibodies (1–2 mg/ml) and IgA-antibodies (around 3 mg/ml) in the serum. IgM-antibodies are the antibodies that are produced following contact with a pathogen. IgA-antibodies are the most important antibodies in mucosal immunity and can be measured via tracheal flushing or tear collection from living animals [4]. The administration of lentogenic ND-live vaccines, such as LaSota or its clone Clone 30, provide local mucosal immunity as well as systemic immunity [23]. Therefore, in this study we chose to analyze the systemic immune response of the tested chickens to the vaccination in terms of IgY-antibodies via blood sampling as the most common testing method for vaccination response in the field [4].

While the common practice of vaccinating backyard poultry chickens every 12 weeks against ND had mostly found acceptance, the Standing Veterinary Committee on Immunization published an announcement on ND-vaccination of backyard poultry that now requires a revaccination every six weeks when using live vaccines [24]. If backyard poultry breeders are forced to vaccinate their animals every six weeks instead of every twelve weeks, there is the risk that backyard poultry will not be vaccinated regularly due to the higher costs and loss of compliance. As a result, the number of immune poultry flocks could significantly decrease and the risk of an outbreak of Newcastle Disease could rise. The aim of our study was to generate field data and evaluate whether the previous vaccination scheme of ND revaccinations every twelve weeks is sufficient or needs to be reconsidered.

## Material and methods

In this study chicken (*Gallus gallus* f. *domestica*) from 27 flocks of backyard poultry breeders in Hesse, Germany were tested. The last vaccination had to have been administered at least 60 days before sampling. Of each flock 30 random blood samples were taken as recommended by Siegmann and Neumann [25]. Usually between 20 and 30 samples are taken as vaccination monitoring [25]. Serology is commonly used to test ND vaccination success in poultry flocks since it is a fast and cost-effective method [4,6,26]. Furthermore, data of all sampled individual chickens such as age (months of life), sex, date of the last vaccination against Newcastle Disease

and total number of vaccinations against Newcastle Disease during lifetime were collected. Selection criteria were solely vaccinations with Nobilis ND Clone 30 (MSD, Munich, Germany) (ND live vaccine) which contains a clone of an ND La Sota strain, or Nobilis Ma 5 + Clone 30 (MSD, Munich, Germany) (IB/ND live vaccine), which also contains a Massachusetts strain of Infectious Bronchitis Virus.

## Vaccination interval in backyard poultry

All tested flocks were vaccinated every 12 weeks with the aforementioned ND and IB/ND live vaccines. Basic immunization schemes according to OIE [27] or producer recommendations [28] were not implemented in the tested flocks as the breeders typically belong to local breeding associations that vaccinate at fixed dates. Moreover, some breeders of a breeding association breed their chicks at different times, so they would have to organize individual vaccinations at different times to achieve a proper basic immunization. Different vaccination schemes are shown in Table 1.

## Blood samples

The blood samples were taken via puncture of the *Vena ulnaris* and collected in 2 ml Microcentrifuge Tubes (Carl Roth GmbH). The samples were cooled for transport, then centrifuged for 3 minutes at 12,000 rpm. The serum was separated from the blood clot and stored at -20°C until use.

## Serological tests

An Enzyme Linked Immunosorbent Assay (ELISA)- test and a Hemagglutination-inhibition test (HI-test) were used to analyze the sera. The HI test is commonly used to test the success of

**Table 1. Different vaccination schemes for chickens\*.**

| Name of organization | Basic immunization | Immunization in laying period |
|---|---|---|
| OIE (ND is mild and sporadic) | 1. doa: Hitchner B1 | not necessary in the first laying period |
| | 18.-21. doa: Hitchner B1 or LaSota | |
| | 10. woa: LaSota | |
| | Point of lay: Inactivated oil vaccine | |
| OIE (ND is severe and widespread) | 1. doa: Hitchner B1 | not necessary in the first laying period |
| | 18.-21. doa: Hitchner B1 or LaSota | |
| | 35.-42. doa: LaSota | |
| | 10. woa: LaSota + Inactivated vaccine | |
| | Point of lay: LaSota + Inactivated vaccine | |
| MSD (for backyard poultry in Germany and Austria) | 3. woa: ND-live vaccine | every 12 months with inactivated vaccine or every 6 to 12 weeks with ND-live vaccine. |
| | 9. woa: ND-live vaccine | |
| | 15. woa: ND-live vaccine | |
| | from the 16. woa: Inactivated vaccine | |
| Vaccination scheme recommended by StIKo Vet | 2.-3. woa ND-live vaccine | every 12 months with inactivated vaccine or every 6 weeks with live vaccines if not vaccinated with inactivated vaccine |
| | 9.-12. woa: ND-live vaccine | |
| | 14.-16. woa: Inactivated vaccine | |
| Vaccination scheme used by breeders in this study | No differentiation between chicks and adult chickens. No classical basic immunization. All chickens (regardless of age) in a flock are vaccinated every 12 weeks with lentogenic ND-live vaccine. | |

\* doa = day of age; woa = week of age.

vaccination because it is relatively cheap whereas the ELISA test is typically used to check for infections because of its high sensitivity [27]. Both tests were performed by the MSD R&D Service Lab (Boxmeer, Netherlands).

To acquire the titer of Anti-NDV-antibodies in the tested chicken serum, a commercial NDV-ELISA from BioChek Immunoassays was used according to the manufacturer's guidelines. The cut-off used was <0.35, i.e. results below this value were scored as negative and non-protective, whereas scores above this cut-off were scored as protective. An HI test was used to examine serum samples for the presence of hemagglutination inhibiting antibodies to Newcastle Disease Virus. Two-fold serial dilutions of the test samples were mixed with an equal volume of NDV antigen. Chicken red blood cells (CRBC) were added and subsequently the dilutions were examined for the presence of complete inhibition of the hemagglutination. According to the OIE, results below $\log_2 4$ are considered as non-protective [27]. Protection against clinical infection and transmission amongst chickens with NDV is given if at least 85% of a flock has a protective titer of at least $\log_2 4$ according to OIE standards [27,29].

## Data collection

To obtain data on the sampled chickens, the breeders were interviewed using a standardized questionnaire. Data collected were age (month of life), breed and sex of the sampled chickens as well as the last date of vaccination of the flock and total number of vaccinations in the lifetime of each chicken. Vaccine data were confirmed by the veterinarian responsible for vaccinating the flock.

## Statistical analysis

The statistical evaluations were made using the statistical program packages BMDP/Dynamic, Release 8.1 [30] and R [31].

To describe and analyze the association between the results of the ELISA and the HI test, a two-dimensional frequency table was built and the number of positive test results were compared with the McNemar test of symmetry. Additionally, the kappa coefficient as a measure of reliability between the methods was computed (all with the program BMDP4F).

To analyze and to quantify the effects of the variables, the impact of vaccine type (VacType), time since the last vaccination (VacDistance), total number of vaccinations (totalVacNo) in their lifetime and breed of the chicken (Breed) (all so-called fixed factors) on the measured titer value of each test system, a partial hierarchical linear mixed effects model (glmm) was fitted to the data using the function lmer from the R library lme4. Due to the high number of different breeds in the flocks, breeds were divided in two classes of to analyze the data: Bantam breeds and normal breeds. The hierarchical ordered random factors were given by the chicken within the flocks. (chicken within flock). In these analyses the following linear model (given in the syntax of the function lmer in the lme4 library of R) was used with the data:

$$\text{ND–titer} \sim \text{VacType} + \text{VacDistance} + \log2(\text{totalNoVac}) + \text{Breed} + (1|\text{Flock})$$

where log2 means the base 2 logarithm, the first four terms of the model equation represent the fixed factors and (1|Flock) the random effects of the chickens within the flocks. The equation was used for the ELISA and the HI test.

Because the statistical distribution of the number of vaccinations was extremely skewed to the right (ranging from 1 to 37), this variable was logarithmically transformed by $\log_2$ in the regression analysis. In all glmm analyses negative titer values were omitted.

**Table 2. Results of the ELISA and HI test.**

| ELISA Test | HI Test | | Total |
|---|---|---|---|
| | **Negative** | **Positive** | |
| **Negative** | 98 (12.3%) | 4 (0.5%) | 102 (12.8%) |
| **Positive** | 103 (13.0%) | 589 (74.2%) | 692 (87.2%) |
| **Total** | 201 (25.3%) | 593 (74.7%) | 794 (100.0%) |

## Results

A total of 810 blood samples were taken for vaccination monitoring from 27 different flocks of backyard poultry breeders with an average flock age of 14 months and a total of 48 different breeds (dwarf and normal breeds of *Gallus gallus* f. *domestica*) of backyard poultry in Hesse, Germany. Seven hundred and ninety-four (794) out of 810 blood samples (98.0%) were analyzed. Sixteen (16) samples could not be analyzed due to insufficient sample size or gelatinization of the sample. The majority of the analyzed samples (696/794 (87.7%)) showed a protective antibody titer against Newcastle Disease based on the ELISA or HI test. The ELISA test showed more positive samples than the HI test (McNemar test: $p < 0.0001$; Table 2 and Fig 1). In total, a value of 0.574 for the kappa coefficient of reliability was found ($p < 0.0001$).

The evaluation of the questionnaires from the breeders showed that 240/794 chickens (30.2%) from eight breeders were vaccinated with the IB/ND-vaccine, whereas 570/794 chickens (71.8%) from 19 breeders were vaccinated with the ND vaccine. The last vaccination was carried out between 69 and 111 days before sampling, with an average of 83.1 days ($\pm$ 9.8 days Standard Deviation) following the last vaccination. Chickens were vaccinated between one and up to 37 times in their lifetime in regular intervals of 12 weeks with an average of 3.25 vaccinations and a median of 1 vaccination per chicken. Chickens tested were between five and 139 months old with a median of 14.4 months. One third of the chickens were male and two-thirds female. The tested chickens belonged to 48 different breeds. These were categorized as

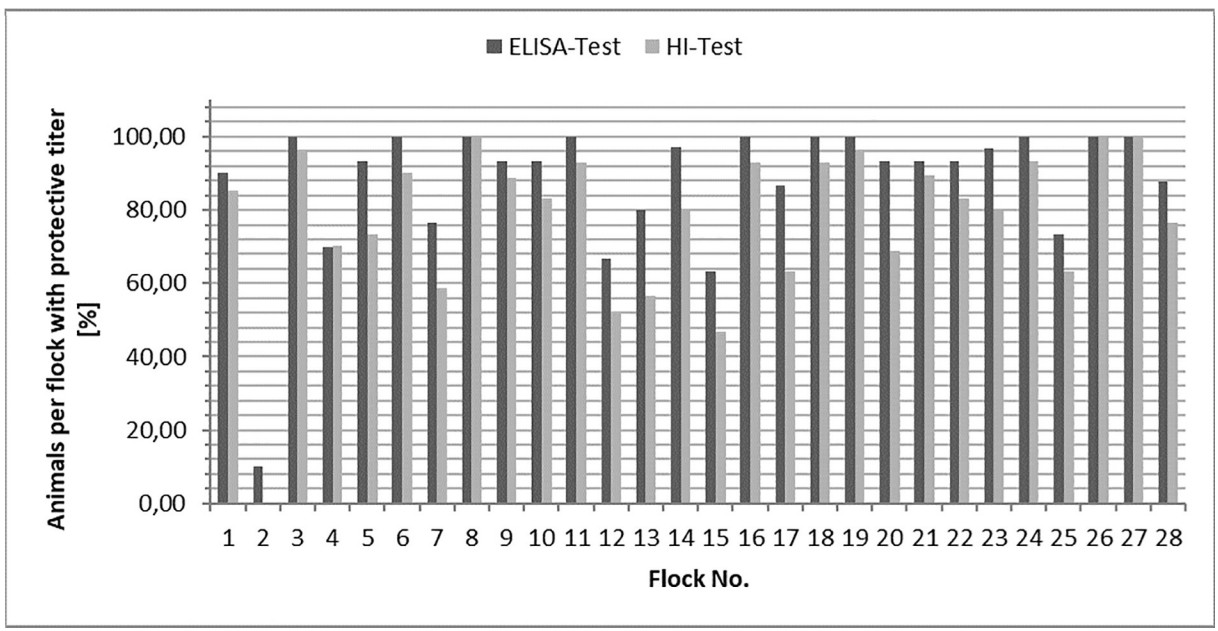

**Fig 1. Proportion of animals with protective titer in ELISA and HI test.**

**Table 3. Mean titer value of each flock in the ELISA test and HI test and percentage of animals with a protective titer per flock\*.**

| Flock No. | Days since last vaccination | ELISA test | HI test | Average age of the tested chickens in months | Percentage of animals per flock with protective titer\*\* (ELISA/HI) |
|---|---|---|---|---|---|
| 1 | 70 | 11.00 | 4.33 | 6.6 | 90.0 (90.0/85.2) |
| 2 | 76 | 1.06 | 0.10 | 5.8 | 10.0 (10.0/0.0) |
| 3 | 69 | 11.27 | 6.13 | 20.6 | 100.0 (100.0/96.6) |
| 4 | 111 | 7.96 | 2.80 | 5.0 | 70.0 (70.0/43.3) |
| 5 | 69 | 11.27 | 5.07 | 9.4 | 93.3 (93.3/73.3) |
| 6 | 87 | 12.24 | 5.10 | 6.2 | 100.0 (100.0/90.0) |
| 7 | 74 | 9.19 | 3.48 (29/30) | 6.2 | 76.7 (76.7/58.6) |
| 8 | 81 | 13.51 | 7.73 | 16.2 | 100.0 (100.0/100.0) |
| 9 | 74 | 11.19 | 4.33 (27/30) | 7.4 | 100.0 (93.3/88.9) |
| 10 | 75 | 10.99 | 4.73 | 5.0 | 93.3 (93.3/83.3) |
| 11 | 90 | 12.39 | 5.82 (28/30) | 13.4 | 100.0 (100.0/93.1) |
| 12 | 79 | 8.15 | 3.52 (29/30) | 13.4 | 66.6 (66.6/51.7) |
| 13 | 72 | 9.55 | 3.69 | 9.0 | 80.0 (80.0/56.7) |
| 14 | 72 | 12.36 | 5.37 | 13.0 | 97.0 (97.0/80.0) |
| 15 | 72 | 7.57 | 2.73 | 7.8 | 63.3 (63.3/46.7) |
| 16 | 87 | 12.35 | 5.68 (28/30) | 10.6 | 100.0 (100.0/92.9) |
| 17 | 87 | 9.50 | 3.47 | 5.0 | 86.6 (83.3/63.3) |
| 18 | 89 | 12.86 | 5.39 (28/30) | 21.0 | 100.0 (100.0/92.9) |
| 19 | 89 | 13.76 | 7.36 (28/30) | 30.2 | 100.0 (100.0/96.4) |
| 20 | 89 | 11.36 | 4.24 (29/30) | 14.6 | 93.3 (93.3/69.0) |
| 21 | 89 | 12.06 | 5.64 (28/30) | 10.6 | 93.3 (93.3/89.3) |
| 22 | 89 | 11.61 | 5.10 | 7.8 | 93.3 (93.3/83.3) |
| 23 | 89 | 11.58 | 4.73 | 6.6 | 96.7 (96.7/80.0) |
| 24 | 89 | 12.71 | 5.87 | 28.6 | 100.0 (100.0/93.3) |
| 25 | 89 | 8.78 | 4.37 | 13.0 | 73.3 (73.3/63.3) |
| 26 | 96 | 12.94 | 7.80 | 31.4 | 100.0 (100.0/100.0) |
| 27 | 90 | 12.22 (28/30) | 6.17 (28/30) | 53.4 | 100.0 (100.0/100.0) |
| total | 83 | 10.86 | 4.84 | 14.0 | 88.0 (87.7/76.7) |

\* (x/x) = number of samples analyzed.

\*\* = chickens that had protective titers in one of both tests regarded as protected.

dwarf (436 chickens of 25 breeds) and normal breeds (374 chickens of 23 breeds) for further investigation.

The mean titer values of all tested flocks were 10.86 in the ELISA test and 4.84 in the HI test (Table 3). The number of animals per flock with a protective titer according to the HI or the ELISA test varied between 10.0% and 100.0% with an average of 88.0%. The results of the ELISA tests exhibited a protective titer (>85% chickens with protective titer) in 19 of 27 flocks. The results of the HI tests showed a protective titer in 13 of 27 flocks. In flock No. 2 only 10.0%

**Table 4. Results of linear regression with the partial hierarchical linear mixed effects regression model.**

**ELISA test**

| | Estimate of the regression coefficient | S.E.[#] of the estimate | t-value | DF | p-value |
|---|---|---|---|---|---|
| (Intercept) | 12.67747 | 0.73100 | 17.343 | 676 | <0.0001 |
| Distance | -0.00966 | 0.00890 | -1.086 | 676 | 0.2781 |
| log$_2$ NumVac* | 0.46227 | 0.03068 | 15.068 | 676 | <0.0001 |
| Vaccine | 0.16749 | 0.19172 | 0.874 | 676 | 0.3826 |
| Breed | -0.11603 | 0.10265 | -1.130 | 676 | 0.2587 |

**HI test**

| | | | | | |
|---|---|---|---|---|---|
| (Intercept) | 5.70464 | 1.52568 | 3.739 | 634 | 0.0002 |
| Distance | -0.01007 | 0.01858 | -0.542 | 634 | 0.5880 |
| log$_2$ NumVac* | 0.82524 | 0.05782 | 14.273 | 634 | <0.0001 |
| Vaccine | -0.06481 | 0.40119 | -0.162 | 634 | 0.8717 |
| Breed | -0.22640 | 0.19366 | -1.169 | 634 | 0.2428 |

* = log$_2$ NumVac = log$_2$ Number of vaccinated animals

[#] = Standard error of the model coefficient estimate.

of the chickens showed a seroconversion with a mean titer value of 1.06 in the ELISA test and 0.10 in the HI test. Both test values are considered non-protective. One flock was vaccinated by dissolving the ND vaccine in cold oatmeal and feeding it directly to the chickens, all chickens of this particular flock showed seroconversion and protective antibody titers.

The analysis of the titer values for both test systems by means of the partial hierarchical linear mixed effects regression model showed only—yet very clearly—a significant effect of the log$_2$ number of vaccinations per chicken (for both tests: p < 0.0001; Table 4). Neither for time since the last vaccination (distance) nor the vaccine nor the breed could statistically significant effects be shown.

For the effect of the log$_2$ number of vaccinations, the regression coefficients allow the interpretation that for the ELISA-titer the mean increase amounts to 0.46 for doubling the number of vaccinations where for the HI test the titer increase equals 0.83 in mean when the number of vaccinations is doubled.

## Discussion

A total of 810 blood samples from 27 flocks of backyard poultry breeders in Hesse, Germany were taken for this study. Of these samples 794 were able to be evaluated. On average, all flocks showed a protective antibody titer in 88.0% (combined test results) of the animals tested. At the time of the last vaccination, the breeders did not know that a serological check would be performed. Due to the voluntary participation of the backyard poultry breeders, it can be assumed that all of the participating breeders are confident in their vaccination technique. This was verified by the positive ND protection results of this study in 21 of 27 cases, where the critical percentage of animals with a protective titer (combined ELISA- and HI-results) of ≥ 85% according to the OIE was reached [27]. To be able to ensure properly performed vaccination, periodic serological controls of the flocks of backyard breeders are recommended. Although all of the samples were taken in Hesse, the results can be transferred to other regions if the vaccination schemes do not differ from the one used in this study (according to Table 1).

All chickens were randomly selected by the breeders giving a cross section of all breeds and ages of the flock. All flocks were clinically healthy. The number of 30 samples taken for vaccination status are the standard number of samples for ND vaccination checks in poultry flocks

[25]. Germany is currently ND free, and to our knowledge there has never been an outbreak of APMV-1 in these flocks. Hence, all detected ND antibodies can be interpreted as vaccination-induced.

Interestingly, flocks that were young in age (average of 5 months) and only vaccinated once, such as flock number 10 (75 days since vaccination) and flock number 17 (87 days since vaccination), already showed sufficient protection based on the combined test results (93.3%/ 86.6%). In literature, it has already been shown that a one-time vaccination with the IB/ND vaccine used in this study can induce a moderate antibody titer which can be verified 40 days post vaccination [32]. It is very likely that the same effect appears if the monovalent ND vaccine is used [33]. The statistical data evaluation showed that the titer of the ND vaccination increased significantly with a doubling of vaccinations. Therefore, it is important to revaccinate chickens properly to booster the immune response until >85% of chickens of a flock have protective titers. This can be achieved through vaccination schemes that propose a basic immunization of the chicks and pullets (e.g. vaccination schemes in Table 1) and a test of the flocks before the laying period. According to the results of this study it is possible to stretch the vaccination interval with ND live vaccines proposed by the StIKo Vet from six to twelve weeks as most of the tested flocks had protective titers (Table 3).

The short revaccination interval of six weeks given in the vaccines' SPCs are presumably owing to the approval procedures of vaccines, requiring cost and time-intensive challenge studies [17,34]. Due to animal welfare issues and costs, it appears plausible to keep the duration of any animal experiment as short as possible. The duration of immunity of vaccines in the EU is usually given as a minimum time span [35]. Furthermore, the largest market for Newcastle Disease live attenuated vaccines is the conventional broiler market, where flocks are slaughtered between four and six weeks of age [25]. In 2017 approx. 677 million broilers were produced in Germany [36]. Broilers are usually vaccinated once with live vaccine via drinking water or spray application [5]. In comparison to these numbers in Germany there are only approximately 300,000 breeding chicken in the sector of backyard poultry from 35,000 breeders according to the Bund Deutscher Rassegeflügelzüchter (BDRG), the largest backyard poultry association worldwide [37]. Due to the aforementioned reasons, there seems to be no commercial need to analyze a potentially longer period of protection, which was shown in this study.

Both test systems, the ELISA and the HI test, were compared in the statistical analysis. The results of both tests were similar, but it was obvious that more samples were positive in the ELISA than in the HI test. It is known that ELISA tests show a higher specificity and sensitivity than HI tests and therefore give more positive results than HI-tests in vaccination checks [38,39]. This can be explained by the nature of both tests. The HI test only detects the antibodies against the HN spike glycoproteins on the surface, whereas ELISA test are potentially able to detect all antibodies against the NDV as the whole virus is used as antigen [40]. In contrast, the ELISA test may detect antibodies which probably do not act as protective antibodies [41]. HI tests are widely used to detect the virus after the occurrence of ND, while ELISA tests are commonly used in vaccination trials. Both tests usually correlate at the level of flock rather than that of the individual animals and can be used to check titers after vaccination of poultry flocks [27]. In light of the broad range of detection of the ND ELISA, we used the test in addition to the HI test as the current gold standard.

There were no significant differences when comparing the use of combined IB/ND vaccine to the use of the ND vaccine. On the one hand this is not surprising, since both vaccines contain the same ND vaccine strain: ND Clone 30. On the other hand, both viruses in the combined vaccine, the IBV and the NDV, target the same cells. Hence, it could be assumed that a simultaneous infection would affect antibody production. Negative effects on antibody

production after a vaccination with mixed monovalent-vaccines against NDV and IBV are well known [42], since the IB-Virus vaccination seems to decrease the capacity of the gland of Harder [43]. This effect does not seem to occur when the combination of both viruses had been carried out by the manufacturer as in case of the IB/ND vaccine used in this study [33]. Interestingly, a study on vaccination against NDV with different vaccines showed that the combined IB/ND vaccine used in this study induces higher antibody production than other combined IB/ND vaccines or an ND vaccine alone after single vaccination. The same study showed that revaccination with a monovalent ND vaccine, the same as used in this study, results in poor antibody titers in comparison to other vaccines [44]. These results could not be confirmed by our study.

It is stated in literature that antibody titers against ND can vary in different breeds or sexes, because of differences of the speed of the metabolism or the stress induced by the onset of laying [45–47]. In this study we tested many different breeds of backyard poultry (46 breeds) as well as some conventional layer hybrids (2 breeds) and hybrids of unknown origin, but there was no significant impact of either sex or breed on the ND antibody titer.

Seven of the 27 flocks showed no sufficient seroconversion in the flocks according to the combined test results. Based on the results of the ELISA test alone 19 flocks showed sufficient seroconversion, while 12 flocks had sufficient seroconversion solely considering the solely the HI test results. The success of a vaccination is influenced by many different factors. On the one hand, the type of chicken (layer or broiler), genetics and age of the birds are biological factors of the chickens themselves. On the other hand, the success of the vaccination is also affected by the vaccination technique. Important aspects of the vaccination technique are the route of vaccination (eye drop, spray, drinking water), the storage of the vaccine, the number of birds per drinking place, the hygiene of the administering vessel, the quality and temperature of the drinking water used to dissolve the vaccine, and the number of doses of vaccine per bird [48]. Common mistakes in vaccinations are poor management with a too long time span between the dissolution and administration of the vaccine, contaminated drinking systems and use of too warm or contaminated water to administer the vaccine. All such mistakes result in a drop of vaccine dose per chicken or even in the administration of completely destroyed virus making the vaccination less or non-effective [49]. Underlying immunosuppressive diseases like Gumboro disease, Marek's disease or Chicken Anemia Virus may also contribute to insufficient antibody-titers [4]. No clinical symptoms of these diseases or higher losses were observed by the breeders in the tested flocks. The reason for insufficient seroconversion in some flocks remains unclear and requires more investigation.

One breeder administered the vaccine for years only through cold oatmeal. Nevertheless, this breeder's flock also showed seroconversion. It has already been shown that some Newcastle Disease vaccination strains could be used to vaccinate chicken via their food such as grain like maize and barley [50,51]. This has not been reported with the ND Clone 30 strain of ND LaSota yet. Furthermore, it is known that oats contain saponins which can influence the production of Newcastle Disease antibodies positively [52,53].

In this study, we showed that a sufficient Newcastle Disease titer ($>$85% according to the OIE) was able to be verified in 20 of 27 backyard poultry flocks with a vaccination interval of twelve weeks. A protective titer which shields the flocks against mortality from Newcastle Disease is highly plausible since another study showed reduced signs of infection and a protection against mortality with higher titers then $\log_2 2$ [29]. The mean number of animals per flock with a protective titer is 88.0% (Table 3), which is sufficient following the minimum requirement of 85% [27,29]. Therefore, we assume that a vaccination interval of twelve weeks can be upheld without sacrificing protection against a Newcastle Disease-outbreak if the flocks are regularly tested and already show a protective titer $>$85%.

## Conclusion and recommendations

This study showed that a vaccination program against Newcastle Disease with a revaccination interval of twelve weeks provides protection against ND for backyard poultry using the Nobilis ND Clone 30 vaccine (MSD, Munich, Germany) or the Nobilis IB Ma5 + ND Clone 30 vaccine (MSD, Munich, Germany) in the flocks studied in Hesse, Germany. The low seroconversion of some flocks emphasizes that flocks have to be tested regularly to detect vaccination failures that can be caused by many factors. The administration of vaccines with the ND Clone 30 strain (MSD, Munich, Germany) through food was sufficient to induce a protective antibody titer in one flock, however further investigations are needed in this regard.

## Supporting information

**S1 Raw data.**
(XLSX)

**S1 File. Statistical analysis.**
(ZIP)

## Acknowledgments

We thank all backyard poultry breeders for participating in and supporting this study by providing their chickens. We thank MSD Animal Health for analyzing the samples in their laboratory. And we also thank Cindy Lea Willheim for proofreading our work.

## Author Contributions

**Conceptualization:** Björn Oberländer, Franca Möller Palau-Ribes.

**Data curation:** Klaus Failing.

**Formal analysis:** Klaus Failing.

**Investigation:** Björn Oberländer, Celina M. Jüngst, Nicole Neuhaus, Franca Möller Palau-Ribes.

**Methodology:** Björn Oberländer, Klaus Failing, Franca Möller Palau-Ribes.

**Project administration:** Björn Oberländer.

**Resources:** Michael Lierz.

**Supervision:** Michael Lierz, Franca Möller Palau-Ribes.

**Validation:** Franca Möller Palau-Ribes.

**Writing – original draft:** Björn Oberländer, Klaus Failing.

**Writing – review & editing:** Klaus Failing, Celina M. Jüngst, Nicole Neuhaus, Michael Lierz, Franca Möller Palau-Ribes.

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
