## [Decision Letter · Decision Letter 0]

9 Jun 2020

PONE-D-20-07821

Evaluation of Newcastle Disease-Antibody titers in backyard poultry in Germany with a vaccination interval of twelve weeks

PLOS ONE

Dear Dr. Oberländer,

Thank you for submitting your manuscript to PLOS ONE. After careful consideration, we feel that it has merit but does not fully meet PLOS ONE’s publication criteria as it currently stands. Therefore, we invite you to submit a revised version of the manuscript that addresses the points raised during the review process.

Manuscript lacks in the quality of preparation. I agree with reviewers, and authors should improve the manuscript. Please review the referee comments and make your peer revision.

We look forward to receiving your revised manuscript.

Kind regards,

Arda Yildirim, Ph.D.

Academic Editor

PLOS ONE

Journal Requirements:

2. We note you have included tables to which you do not refer in the text of your manuscript. Please ensure that you refer to Tables 1-3 in your text; if accepted, production will need this reference to link the reader to the Table.

Additional Editor Comments (if provided):

This manuscript is well-designed work; however there is a major flaw in the interpretation of the data methodology, improving the clarity of arguments, English language and redaction style. It is necessary to improve the manuscript by examining the questions that need to be clarified in a way. For your guidance, you can check the reviewers' comments. Thank you for giving us the opportunity to consider your work.

Reviewers' comments:

Reviewer's Responses to Questions

**Comments to the Author**

1. Is the manuscript technically sound, and do the data support the conclusions?

Reviewer #1: Partly

Reviewer #2: Partly

Reviewer #3: Yes

Reviewer #4: Partly

2. Has the statistical analysis been performed appropriately and rigorously? 

Reviewer #1: Yes

Reviewer #2: Yes

Reviewer #3: Yes

Reviewer #4: Yes

3. Have the authors made all data underlying the findings in their manuscript fully available?

Reviewer #1: Yes

Reviewer #2: Yes

Reviewer #3: Yes

Reviewer #4: Yes

4. Is the manuscript presented in an intelligible fashion and written in standard English?

Reviewer #1: No

Reviewer #2: Yes

Reviewer #3: Yes

Reviewer #4: Yes

5. Review Comments to the Author

Reviewer #1: Overall, there is a lot of value to this manuscript in determining NDV antibody titers after vaccination in backyard flocks for prolonged periods (69 – 111 days), as understanding titer levels for longer durations is critical to understanding protection for backyard flocks versus commercial flocks. Backyard flocks can be major reservoirs for many different poultry diseases in numerous countries if not vaccinated and maintained correctly, thus understanding these parameters to help in defining the most accurate recommendations for backyard breeders is vital to global poultry health. However, the major issue with this manuscript is the discussion section, as there are numerous problems including confusing statements and drawing conclusions not supported by the results of the study. The discussion section should be completely rewritten to better discuss the specific results of the study, and what they mean to the bigger picture including the current literature. Additionally, the manuscript overall needs to be polished up and the writing cleaned up to make the general message clearer to the reader and easier to understand in general. There are many grammatical errors and typos throughout the manuscript that should be corrected by the authors.

Specific comments:

Abstract

Line 27 – 32 – It is unclear to the initial reader is the purpose to compare 6 weeks recommended by the Standing Committee or SPCs at 12 weeks. Later it becomes clear that SPCs also recommend the 6 weeks, but the standard practice in Germany is 12 weeks. Therefore, these two sentences should be re-worked to clarify the meaning.

Introduction

Line 47 – 51 – Order of these sentences flows very weird and is a little confusing to the reader.

Line 52 – 59 – General paragraph feels like just random facts put together. Might be worth defining velogenic, mesogenic, lentogenic, and asymptomatic enteric strains a little better. Additionally, might be worth mentioning that lentogenic viral strains are commonly used for vaccines.

Line 62 – Beginning of sentence does not flow with previous sentence, confusing

Line 63 – poultry infection should be changed to poultry outbreak

Line 66 – poultry is vaccinated to poultry are vaccinated

Line 80 – 84 – Run-on sentence, should be broken up

Line 100 – 102 – Cell mediated immunity is the immune response that does not involve antibodies. Plus, mucosal immunity would suggest IgA response, should be mentioned how serum levels of IgY correspond to IgA levels for protection. This sentence is a little confusing and should be clarified

Materials and Methods

Line 120 – Hesse, Germany

Line 130 – 134 – How close is the blood collection method to that commonly used for antibody titer checks in flocks? This should be mentioned and/or referenced.

Line 138 – control changed to test

Line 139 – ELISA-tet to ELISA-test

Line 142 – chickenserum to chicken serum

Line 146 – An HI-Test to A HI-Test

Line 155 – Line 157 – confusing is this saying flock immunity can vary or % protective titer of the flock, needs clarification

Results

Table 2 – Might be worth including breeds within the flocks and/or average age of flocks in this table

Table 2 – correct the switch between commas and decimal points in the numbers

Line 218 – There is no table 4 as cited in the paper

Line 216 – 220 – This section should be re-worked and cleaned up for clarity and grammatical errors

Discussion

Since the # of vaccines had the biggest statistical impact on the antibody titers would this not argue for more frequent vaccinations, thus every 6 weeks instead of 12 weeks. However, this is not discussed in the discussion, and the major statistical factor determined in the study is not even really mentioned in the discussion of the paper. This should be addressed

Line 233 – Hesse, Germany

Line 234 – 241 – The authors keep switching between the 58% and the OIE recommended 85% for flock immunity when it seems to fit their discussion better. What does the German Veterinarian Standing Committee consider the %? Would be better throughout the paper to stick with one percentage that has been justified and is used in Germany.

Line 242 – 244 – Seems unlikely that the authors are going to find a region with the exact same vaccines, vaccination intervals, breeds, and climatic parameters for this to perfectly match as stated in the sentence. Additionally, the results of the paper indicate that there is no statistical impact due to vaccine type, last vaccine intervals or breeds therefore the authors are arguing against the authors’ own results. Need to comment on the author’s own results and the impact beyond Hesse, Germany.

Line 245 – chicken to chickens

Line 247 – 30 samples taken for the vaccination control were determined statistically? There is no mention in the methods about running statistics to determine the 30 samples

Line 251 – 252 – No data is present in the paper to support this statement, there was only one time point taken from the chickens therefore the authors cannot compare between flocks to say there is no decrease in titers between 69 and 111 days after vaccination. The authors would need to study the flocks at different time points to determine that within each flock there was not a drop across those time periods, because it is possible that the flocks at 111 days had an extremely high titer at 69 days that has dropped down by 111 days.

Line 254 – are dropping to drop

Line 257 – to literature change to the literature

Line 259 – 261 – statement of doubling the vaccination would again argue in favor of every 6 weeks instead of 12 weeks, which would double the vaccination

Line 278 – 281 – I understand the reason for the information, but these comments seem to just be dropped into the end of the paragraph and do not flow with the rest of the paragraph.

Line 282 – 284 – This sentence contradicts itself, if they are congruent then they are the same. Additionally, the kappa coefficient of reliability was 0.574 that could be argued to around the middle ground, so this should be discussed more in the discussion and the impact ELISA versus HI-Test could have on predicting flock immunity. The authors need to discuss their own results and the meaning.

Line 296 – 298 – unclear sentence

Line 298 – 303 – Unclear as to the point of this section and how it fits into the results of the study, particularly as the results show no statistical difference in type of vaccine

Line 306 – 314 – How does this section relate to the breeds in this study? This section seems to go off on a tangent to just fill space, discuss the results of the paper

Line 327 – 329 – No evidence this is what happened, particularly as the authors just mentioned numerous other factors that could impact the antibody titer in a flock. Was this flock older? Or younger? Different breed compared to all the others? Additionally, the authors present no data to suggest there was any issues with vaccination technique.

Line 332 – 333 – Sentence needs to be fixed

Line 339 – 341 – sentence is confusing OIE was 85% whereas again using 58% to justify the discussion. Need to justify one percentage and use it throughout the manuscript particularly the discussion.

Line 344 – 346 – Confusing sentence, needs fixing

Conclusion

Line 354 – 356 – Again, the authors do not present any data to be able to draw this conclusion on the handling of the vaccine. Either data needs to be added to the manuscript to justify the statement or remove the statement.

Reviewer #2: 1. The meaning of this report on the evaluate if a vaccination interval of twelve weeks against Newcastle Disease under field conditions results in sufficient seroconversion to protect the flocks is significant, especially in Germany. By the author’s work, some useful results gotten, like a vaccination interval of every twelve weeks with the live-vaccines tested is suitable for a vaccination protocol against Newcastle Disease..

2. Vaccine can cause an enormous change of immunocytes,cytokines and antibody, cause a series of pathological and physiological change in the body.Therefore I suggest the author could discuss deeper about the effection about the vacction protocol by whether NDV infection rate were reduced which can be diagnosed by a number of different laboratory tests.For example,conclusion the change of B cell and T cell can reflected the pathogenesis. It may improve the diagnose. The main support data were only serological testing results I thought not enough.

3. If the author showed the data by graphs will be better than too many tables.

Reviewer #3: The manuscript presents relevant data for poultry practitioners and producers in Germany.

I have just a few observations as follows;

Since it has been stated that there is a current directive to vaccinate birds at 6 weeks, what necessitated that decision? Was is based on a study recommendation?

Is the protective antibody observed in this study significantly higher that what was observed following vaccination at 6 weeks?

Some of the flocks used in the study had less than 12 weeks since the last vaccination, would it not be better to give a range in days instead of weeks ?

Again, in order to appreciate the lower and upper limits, it would be preferable to use Mean and SE or SD.

In-text citations are inconsistent e,g lines 248, 270,281, 286, 314, 322....just to mention a few. Use standard journal citation and referencing style.

Reviewer #4: [General comments]

This paper describes the analytical results on a survey for anti-NDV antibodies to discuss whether twelve-week interval of NDV vaccination is suffice to keep backyard poultry in Germany. The authors concluded that a vaccination interval of twelve weeks with the live-vaccines was suitable for vaccination protocol in backyard poultry.

The authors describes as the reason for this study is the direction from German Veterinarian Standing Committee of Immunization to re-vaccinate for NDV every six weeks, instead of every twelve weeks, which was regularly performed in Germany. However, in OIE manual issued in 2006, it is recommended to vaccinate every 8 weeks to keep HI titer of 2^5. In this manuscript, the reason for the change to re-vaccination of six-week interval was not clearly described, thus, for the reviewer, the contents of this paper seemed to be of domestic importance.

[[Comments for the contents]]

[Sample information]

Because the samples were taken from backyard chichen, the age, breed, aim and so on, would be variable. However, as the focus of this paper is to demonstrate the comparable efficacy to keep anti-NDV antibody titer by re-vaccination in twelve-week intervals, compared to those in six-week intervals.

In the S1 Raw data sheet, flock number, age in month, number of times of vaccination, and ELISA and HI for titersindividuals are listed. In the text, the period from the last vaccination is described as 69-111days (approx. 10-16 weeks after LAST vaccination). However, I could not understand in which flock or even in which individuals followed six- or twelve-week interval revaccination, if the repeats of vaccination affects to the remaining titer. For example, chickens of 29 weeks old in different flock received 7 vaccinations, while 5 months chickens which are the major age group in this study encompassing 490 chicken out of total 810 chickens (60.5%) received 1 time vaccination. In OIE manual (2006), breeding hens are recommended to vaccinate twice by 14 or 18 weeks of age, for layer and broiler chickens, respectively (ND3.1-16FINAL(29Nov06) appendix 5). So, the more than half of chickens of 5 month received only single vaccination means that their vaccination schedule was different from the regular ones. The detailed schedule of vaccination of backyard chickens should be noticed for better understanding of the situdation.

[Value in tables]

From the reference of van Boven et al (2008) demonstrating vaccinated chickens with HI titers of 2^3 or more with strong protection, setting HI titer of 2^4 as the minimum protective titer would be acceptable. However, in materials and methods, the authors described that HI titer of 2^3 or less is considered as negative (according to OIE). I think it is unclear statement because HI titer of 2-1 to 2-3 is still clear in HI titer. OIE manual explained that as a protective titer, they recommended to determine HI titer of 2^4 or 2^3.

In table 2, the flock 1 including 30 samples in which includes 3 unmeasulable samples, 3 samples with HI titer of <1, one of HI titer of 3. That means "protective" titer of 2^4 or more would be 23 out of 27 available sample, equals to 85.2%, which is described as 90% protective. As similar, in flock 2, all of 30 samples showed HI titers of 1 or <1. So, percentage of protective titer should be 0.0% (0/30), which is described as 10%. In materials and methods, the authors mentioned that they used R software. However, the authors should describe essential calculation formulars in the materials and methods so that the readers can easily understand.

In table 2 "," and "." are mixed as period. "." should be used.

As similar, protective HI titer in table 2 is 89%, while positive HI test (2^4 or more?) in table 1 is 74.7%. My calculation from supplimented law data for proportion of HI titers 2^4 or more is 75.3% (593/788: Total 810, IS=22, <1=122, 1-3=73). No formular or legends (because they are tables) was indicated for the way of calculation.

[Result interpretation]

The title of this manuscript implies to demonstrate the decline of HI titer (or additionally ELISA titer for higher sensitivity) is calm (slow) enough to maintains the anti-NDV antibody titers at the protective levels. Results demonstrated high HI titers were still detected after 12 weeks (84 days). As described in the discussion, the factors such as the way of immunization, breeds, and others might affect to the (maintainance of) HI titers, description of the schedule of vaccination in addition to the number of repeated vaccination times would be required, as described above.

[Correlation between ELISA titer and HI titer]

Generally, neutralization titers are not always linearly correlated to whole antibody titers. In some our experiences, the protortion of neutralizing titer increased after repeated immunization. I think the protective value should be based on the HI titer not ELISA titer. In this section, more clear explanation about the aim of statistical comparison of these two values in order to evaluate the comparable efficacy of re-vaccination in twelve-week intervals.

[breeds, ages, flock differences]

Even no significant differences were demonstrated, the expectation or some examples in references which shows the difference between dofferent categories. Or if the authors put them for evaluation in case there might have some unexpected differences between breeds or others, the authors should not conclude that those differences might affect to the results (lines 315-322).

[Proofreading and other]

Some grammatical mistakes and mistypes were found.　Order of the significant digits for the same category of values should be same.

6. PLOS authors have the option to publish the peer review history of their article (what does this mean?). If published, this will include your full peer review and any attached files.

Reviewer #1: No

Reviewer #2: Yes: Hao Peng

Reviewer #3: No

Reviewer #4: No

---

## [Author Response · Author response to Decision Letter 0]

23 Jul 2020

We addressed all issues raised by the reviewers as follows:

Review Comments to the Author

Reviewer #1: Overall, there is a lot of value to this manuscript in determining NDV antibody titers after vaccination in backyard flocks for prolonged periods (69 – 111 days), as understanding titer levels for longer durations is critical to understanding protection for backyard flocks versus commercial flocks. Backyard flocks can be major reservoirs for many different poultry diseases in numerous countries if not vaccinated and maintained correctly, thus understanding these parameters to help in defining the most accurate recommendations for backyard breeders is vital to global poultry health. However, the major issue with this manuscript is the discussion section, as there are numerous problems including confusing statements and drawing conclusions not supported by the results of the study. The discussion section should be completely rewritten to better discuss the specific results of the study, and what they mean to the bigger picture including the current literature. Additionally, the manuscript overall needs to be polished up and the writing cleaned up to make the general message clearer to the reader and easier to understand in general. There are many grammatical errors and typos throughout the manuscript that should be corrected by the authors.

• The authors thank the reviewer for the comments and suggestions. The authors rewrote the discussion according to the reviewer’s suggestions and we had the study proof-read by a native speaker. The authors addressed the comments on a point-to-point basis as follows:

Specific comments:

Abstract

Line 27 – 32 – It is unclear to the initial reader is the purpose to compare 6 weeks recommended by the Standing Committee or SPCs at 12 weeks. Later it becomes clear that SPCs also recommend the 6 weeks, but the standard practice in Germany is 12 weeks. Therefore, these two sentences should be re-worked to clarify the meaning.

• We rewrote the sentences, to clarify the meaning.

Introduction

Line 47 – 51 – Order of these sentences flows very weird and is a little confusing to the reader.

• We rewrote/reorganized the sentence to get a better flow and clarify information.

Line 52 – 59 – General paragraph feels like just random facts put together. Might be worth defining velogenic, mesogenic, lentogenic, and asymptomatic enteric strains a little better. Additionally, might be worth mentioning that lentogenic viral strains are commonly used for vaccines.

• We rewrote the paragraph to provide the facts in a clearer way. We defined the different strains of ND and also mentioned the lentogenic ND strains later on.

Line 62 – Beginning of sentence does not flow with previous sentence, confusing

• We rewrote the section.

Line 63 – poultry infection should be changed to poultry outbreak

• This was amended as suggested.

Line 66 – poultry is vaccinated to poultry are vaccinated

• This was amended as suggested.

Line 80 – 84 – Run-on sentence, should be broken up

• This was amended as suggested.

Line 100 – 102 – Cell mediated immunity is the immune response that does not involve antibodies. Plus, mucosal immunity would suggest IgA response, should be mentioned how serum levels of IgY correspond to IgA levels for protection. This sentence is a little confusing and should be clarified

• We rewrote the section to clarify the immune response and also included information about the relation of IgY and IgA.

Materials and Methods

Line 120 – Hesse, Germany

• This was amended as suggested.

Line 130 – 134 – How close is the blood collection method to that commonly used for antibody titer checks in flocks? This should be mentioned and/or referenced.

• This is the common method to test antibody titers in flocks. We made this clear in the revised manuscript and referred to literature in this regard.

Line 138 – control changed to test

• This was amended as suggested.

Line 139 – ELISA-tet to ELISA-test

• This was amended as suggested.

Line 142 – chickenserum to chicken serum

• This was amended as suggested.

Line 146 – An HI-Test to A HI-Test

• This was amended as suggested.

Line 155 – Line 157 – confusing is this saying flock immunity can vary or % protective titer of the flock, needs clarification

• We clarified this by sticking to the OIE recommendations throughout the whole manuscript.

Results

Table 2 – Might be worth including breeds within the flocks and/or average age of flocks in this table

• We included the average age of the flocks in the table. When we tried to include the different breeds, the resulting table was confusing.

Table 2 – correct the switch between commas and decimal points in the numbers

• This was amended as suggested.

Line 218 – There is no table 4 as cited in the paper

• We corrected the information regarding the tables and cited them properly in the revised manuscript.

Line 216 – 220 – This section should be re-worked and cleaned up for clarity and grammatical errors

• We rewrote the section as suggested.

Discussion

Since the # of vaccines had the biggest statistical impact on the antibody titers would this not argue for more frequent vaccinations, thus every 6 weeks instead of 12 weeks. However, this is not discussed in the discussion, and the major statistical factor determined in the study is not even really mentioned in the discussion of the paper. This should be addressed

• We tried to address this more clearly in the discussion. In our opinion based on our filed observations a shorter vaccination interval reduces the willingness of the breeders to vaccinate, so the interval of 12 weeks seems to be a good compromise to uphold the “ND-free”-status of Germany.

Line 233 – Hesse, Germany

• This was amended as suggested.

Line 234 – 241 – The authors keep switching between the 58% and the OIE recommended 85% for flock immunity when it seems to fit their discussion better. What does the German Veterinarian Standing Committee consider the %? Would be better throughout the paper to stick with one percentage that has been justified and is used in Germany.

• Now we stick to the recommendations of the OIE (85 %) throughout the whole paper. The StIKo Vet also consider the 85 % of the OIE as standard.

Line 242 – 244 – Seems unlikely that the authors are going to find a region with the exact same vaccines, vaccination intervals, breeds, and climatic parameters for this to perfectly match as stated in the sentence. Additionally, the results of the paper indicate that there is no statistical impact due to vaccine type, last vaccine intervals or breeds therefore the authors are arguing against the authors’ own results. Need to comment on the author’s own results and the impact beyond Hesse, Germany.

• We rewrote the section as the reviewer suggested.

Line 245 – chicken to chickens

• This was amended as suggested.

Line 247 – 30 samples taken for the vaccination control were determined statistically? There is no mention in the methods about running statistics to determine the 30 samples

• We rewrote the sentence to clarify this point. The number of samples were based on literature research.

Line 251 – 252 – No data is present in the paper to support this statement, there was only one time point taken from the chickens therefore the authors cannot compare between flocks to say there is no decrease in titers between 69 and 111 days after vaccination. The authors would need to study the flocks at different time points to determine that within each flock there was not a drop across those time periods, because it is possible that the flocks at 111 days had an extremely high titer at 69 days that has dropped down by 111 days.

• We removed the statement as suggested by the reviewer.

Line 254 – are dropping to drop

• This was amended as suggested.

Line 257 – to literature change to the literature

• This was amended as suggested.

Line 259 – 261 – statement of doubling the vaccination would again argue in favor of every 6 weeks instead of 12 weeks, which would double the vaccination

• We rewrote the section to clarify that the interval of twelve weeks could be a compromise to uphold flock immunity and willingness to vaccinate properly.

Line 278 – 281 – I understand the reason for the information, but these comments seem to just be dropped into the end of the paragraph and do not flow with the rest of the paragraph.

• We rewrote the section to include the information in a more meaningful way.

Line 282 – 284 – This sentence contradicts itself, if they are congruent then they are the same. Additionally, the kappa coefficient of reliability was 0.574 that could be argued to around the middle ground, so this should be discussed more in the discussion and the impact ELISA versus HI-Test could have on predicting flock immunity. The authors need to discuss their own results and the meaning.

• We rewrote the section and tried to clarify the impact of ELISA vs HI test according to flock immunity.

Line 296 – 298 – unclear sentence

Line 298 – 303 – Unclear as to the point of this section and how it fits into the results of the study, particularly as the results show no statistical difference in type of vaccine

Line 306 – 314 – How does this section relate to the breeds in this study? This section seems to go off on a tangent to just fill space, discuss the results of the paper

• We rewrote the section, deleted unnecessary information and sticked to the results of the paper.

Line 327 – 329 – No evidence this is what happened, particularly as the authors just mentioned numerous other factors that could impact the antibody titer in a flock. Was this flock older? Or younger? Different breed compared to all the others? Additionally, the authors present no data to suggest there was any issues with vaccination technique.

• To our experience issues with the vaccination technique are the most common problems in vaccination backyard poultry flocks. Since we cannot prove that this was the cause in this case, we removed the statement.

Line 332 – 333 – Sentence needs to be fixed

• We rewrote the section and fixed all sentences.

Line 339 – 341 – sentence is confusing OIE was 85% whereas again using 58% to justify the discussion. Need to justify one percentage and use it throughout the manuscript particularly the discussion.

• We rewrote the paper and now stick to the OIE-recommendations throughout the whole paper.

Line 344 – 346 – Confusing sentence, needs fixing

• We rewrote the section and fixed all sentences.

Conclusion

Line 354 – 356 – Again, the authors do not present any data to be able to draw this conclusion on the handling of the vaccine. Either data needs to be added to the manuscript to justify the statement or remove the statement.

• We rewrote the section and removed the statement.

Reviewer #2: 1. The meaning of this report on the evaluate if a vaccination interval of twelve weeks against Newcastle Disease under field conditions results in sufficient seroconversion to protect the flocks is significant, especially in Germany. By the author’s work, some useful results gotten, like a vaccination interval of every twelve weeks with the live-vaccines tested is suitable for a vaccination protocol against Newcastle Disease.

• The authors thank the reviewer for his motivating comments.

2. Vaccine can cause an enormous change of immunocytes,cytokines and antibody, cause a series of pathological and physiological change in the body.Therefore I suggest the author could discuss deeper about the effection about the vacction protocol by whether NDV infection rate were reduced which can be diagnosed by a number of different laboratory tests.For example,conclusion the change of B cell and T cell can reflected the pathogenesis. It may improve the diagnose. The main support data were only serological testing results I thought not enough.

• The authors thank the reviewer for his suggestions. Testing the changes of B and T cells is an interesting option that we bear in mind for further studies. Due to the value of each chicken to the breeder the collection of blood samples from living chickens was the only option available. Serological testing is also the standard choice of measurement of the vaccination success in commercial poultry flocks.

3. If the author showed the data by graphs will be better than too many tables.

• We included one graph on the proportion of animals with protective titer in ELISA and HI test.

Reviewer #3: The manuscript presents relevant data for poultry practitioners and producers in Germany.

I have just a few observations as follows;

• The authors thank the reviewer for his observations.

Since it has been stated that there is a current directive to vaccinate birds at 6 weeks, what necessitated that decision? Was is based on a study recommendation?

• The current directive is based on the current SPCs of available ND vaccines. 

Is the protective antibody observed in this study significantly higher that what was observed following vaccination at 6 weeks?

• Unfortunately, we have no data regarding the antibody titer at 6 weeks after the vaccination in backyard poultry. We will keep this in mind for further studies.

Some of the flocks used in the study had less than 12 weeks since the last vaccination, would it not be better to give a range in days instead of weeks ?

• Since it is common in Germany to stick to weeks as interval for vaccination in poultry (backyard as well as commercial), we would prefer to give our recommendation in weeks to make it easier to understand for the reader.

Again, in order to appreciate the lower and upper limits, it would be preferable to use Mean and SE or SD.

• We specified the SD at the vaccination distance. In case of the number of the vaccinations the SD was not meaningful due to the extremely skewed distribution of the values to the right.

In-text citations are inconsistent e,g lines 248, 270,281, 286, 314, 322....just to mention a few. Use standard journal citation and referencing style.

• We changed the citation to Vancouver-Style as outlined in the journal guidelines.

Reviewer #4: [General comments]

This paper describes the analytical results on a survey for anti-NDV antibodies to discuss whether twelve-week interval of NDV vaccination is suffice to keep backyard poultry in Germany. The authors concluded that a vaccination interval of twelve weeks with the live-vaccines was suitable for vaccination protocol in backyard poultry.

The authors describes as the reason for this study is the direction from German Veterinarian Standing Committee of Immunization to re-vaccinate for NDV every six weeks, instead of every twelve weeks, which was regularly performed in Germany. However, in OIE manual issued in 2006, it is recommended to vaccinate every 8 weeks to keep HI titer of 2^5. In this manuscript, the reason for the change to re-vaccination of six-week interval was not clearly described, thus, for the reviewer, the contents of this paper seemed to be of domestic importance.

• The authors thank the reviewer for his comments and suggestions. We rewrote some sections of the paper and clarify the reason for the change of the revaccination interval. 

[[Comments for the contents]]

[Sample information]

Because the samples were taken from backyard chichen, the age, breed, aim and so on, would be variable. However, as the focus of this paper is to demonstrate the comparable efficacy to keep anti-NDV antibody titer by re-vaccination in twelve-week intervals, compared to those in six-week intervals.

In the S1 Raw data sheet, flock number, age in month, number of times of vaccination, and ELISA and HI for titers individuals are listed. In the text, the period from the last vaccination is described as 69-111days (approx. 10-16 weeks after LAST vaccination). However, I could not understand in which flock or even in which individuals followed six- or twelve-week interval revaccination, if the repeats of vaccination affects to the remaining titer. For example, chickens of 29 weeks old in different flock received 7 vaccinations, while 5 months chickens which are the major age group in this study encompassing 490 chicken out of total 810 chickens (60.5%) received 1 time vaccination. In OIE manual (2006), breeding hens are recommended to vaccinate twice by 14 or 18 weeks of age, for layer and broiler chickens, respectively (ND3.1-16FINAL(29Nov06) appendix 5). So, the more than half of chickens of 5 month received only single vaccination means that their vaccination schedule was different from the regular ones. The detailed schedule of vaccination of backyard chickens should be noticed for better understanding of the situdation.

• All flocks in this study were vaccinated in a twelve-week interval only. We added a table with different vaccination schemes including recommendations for basic immunization to clarify the actual situation in backyard poultry and to show the differences according to the recommendations for vaccination.

[Value in tables]

From the reference of van Boven et al (2008) demonstrating vaccinated chickens with HI titers of 2^3 or more with strong protection, setting HI titer of 2^4 as the minimum protective titer would be acceptable. However, in materials and methods, the authors described that HI titer of 2^3 or less is considered as negative (according to OIE). I think it is unclear statement because HI titer of 2-1 to 2-3 is still clear in HI titer. OIE manual explained that as a protective titer, they recommended to determine HI titer of 2^4 or 2^3.

• We changed the wording from “negative” to “non-protective”.

In table 2, the flock 1 including 30 samples in which includes 3 unmeasulable samples, 3 samples with HI titer of <1, one of HI titer of 3. That means "protective" titer of 2^4 or more would be 23 out of 27 available sample, equals to 85.2%, which is described as 90% protective. As similar, in flock 2, all of 30 samples showed HI titers of 1 or <1. So, percentage of protective titer should be 0.0% (0/30), which is described as 10%. In materials and methods, the authors mentioned that they used R software. However, the authors should describe essential calculation formulars in the materials and methods so that the readers can easily understand.

• We reevaluated and corrected the numbers and also rewrote the section to clarify the way of calculation.

In table 2 "," and "." are mixed as period. "." should be used.

• This was amended as suggested.

As similar, protective HI titer in table 2 is 89%, while positive HI test (2^4 or more?) in table 1 is 74.7%. My calculation from supplimented law data for proportion of HI titers 2^4 or more is 75.3% (593/788: Total 810, IS=22, <1=122, 1-3=73). No formular or legends (because they are tables) was indicated for the way of calculation.

• Also for the HI-titers we recalculated the numbers included in the table and also rewrote the section to clarify how we calculated the results.

[Result interpretation]

The title of this manuscript implies to demonstrate the decline of HI titer (or additionally ELISA titer for higher sensitivity) is calm (slow) enough to maintains the anti-NDV antibody titers at the protective levels. Results demonstrated high HI titers were still detected after 12 weeks (84 days). As described in the discussion, the factors such as the way of immunization, breeds, and others might affect to the (maintainance of) HI titers, description of the schedule of vaccination in addition to the number of repeated vaccination times would be required, as described above.

• As stated above, we included a table with the vaccination scheme used by the breeders whose chickens were tested in this study.

[Correlation between ELISA titer and HI titer]

Generally, neutralization titers are not always linearly correlated to whole antibody titers. In some our experiences, the protortion of neutralizing titer increased after repeated immunization. I think the protective value should be based on the HI titer not ELISA titer. In this section, more clear explanation about the aim of statistical comparison of these two values in order to evaluate the comparable efficacy of re-vaccination in twelve-week intervals.

• We rewrote the section about ELISA and HI titer in the discussion to show the differences of both tests. For this study we decided to combine the results of both tests to specify the protective titer.

[breeds, ages, flock differences]

Even no significant differences were demonstrated, the expectation or some examples in references which shows the difference between dofferent categories. Or if the authors put them for evaluation in case there might have some unexpected differences between breeds or others, the authors should not conclude that those differences might affect to the results (lines 315-322).

• We removed all statements that could argue against the results in this study.

[Proofreading and other]

Some grammatical mistakes and mistypes were found.　Order of the significant digits for the same category of values should be same.

• We correct the grammatical mistakes as well as the mistypes.

---

## [Decision Letter · Decision Letter 1]

10 Aug 2020

Evaluation of Newcastle Disease Antibody titers in backyard poultry in Germany with a vaccination interval of twelve weeks

PONE-D-20-07821R1

Dear Dr. Oberländer,

We’re pleased to inform you that your manuscript has been judged scientifically suitable for publication and will be formally accepted for publication once it meets all outstanding technical requirements.

Kind regards,

Arda Yildirim, Ph.D.

Academic Editor

PLOS ONE

Additional Editor Comments (optional):

Thanks for your hard work.

Reviewers' comments:

Reviewer's Responses to Questions

**Comments to the Author**

1. If the authors have adequately addressed your comments raised in a previous round of review and you feel that this manuscript is now acceptable for publication, you may indicate that here to bypass the “Comments to the Author” section, enter your conflict of interest statement in the “Confidential to Editor” section, and submit your "Accept" recommendation.

Reviewer #2: All comments have been addressed

Reviewer #3: All comments have been addressed

2. Is the manuscript technically sound, and do the data support the conclusions?

Reviewer #2: Partly

Reviewer #3: Yes

3. Has the statistical analysis been performed appropriately and rigorously? 

Reviewer #2: Yes

Reviewer #3: Yes

4. Have the authors made all data underlying the findings in their manuscript fully available?

Reviewer #2: (No Response)

Reviewer #3: Yes

5. Is the manuscript presented in an intelligible fashion and written in standard English?

Reviewer #2: Yes

Reviewer #3: Yes

6. Review Comments to the Author

Reviewer #2: The author rewrote the discussion according to the reviewer’s suggestions.For me, I can accept the revised paper.

Reviewer #3: The authors have addressed all relevant comments raised in the manuscript. The manuscript can be accepted for publication.

7. PLOS authors have the option to publish the peer review history of their article (what does this mean?). If published, this will include your full peer review and any attached files.

Reviewer #2: **Yes: **Hao Peng

Reviewer #3: **Yes: **Dr. Yusuf Abba

---

## [Editor Report · Acceptance letter]

14 Aug 2020

PONE-D-20-07821R1 

Evaluation of Newcastle Disease Antibody titers in backyard poultry in Germany with a vaccination interval of twelve weeks 

Dear Dr. Oberländer:

I'm pleased to inform you that your manuscript has been deemed suitable for publication in PLOS ONE. Congratulations! Your manuscript is now with our production department. 

Kind regards, 

on behalf of

Dr. Arda Yildirim 

Academic Editor

PLOS ONE